# econSG: Efficient and Multi-view Consistent Open-Vocabulary 3D Semantic Gaussians

**Can Zhang & Gim Hee Lee**
Department of Computer Science
National University of Singapore
`can.zhang@u.nus.edu, gimhee.lee@nus.edu.sg`

## Abstract

The primary focus of most recent works on open-vocabulary neural fields is extracting precise semantic features from the VLMs and then consolidating them efficiently into a multi-view consistent 3D neural fields representation. However, most existing works over-trusted SAM to regularize image-level CLIP without any further refinement. Moreover, several existing works improved efficiency by dimensionality reduction of semantic features from 2D VLMs before fusing with 3DGS semantic fields, which inevitably leads to multi-view inconsistency. In this work, we propose econSG for open-vocabulary semantic segmentation with 3DGS. Our econSG consists of: 1) A Confidence-region Guided Regularization (CRR) that mutually refines SAM and CLIP to get the best of both worlds for precise semantic features with complete and precise boundaries. 2) A low dimensional contextual space to enforce 3D multi-view consistency while improving computational efficiency by fusing backprojected multi-view 2D features and follow by dimensional reduction directly on the fused 3D features instead of operating on each 2D view separately. Our econSG shows state-of-the-art performance on four benchmark datasets compared to the existing methods. Furthermore, we are also the most efficient training among all the methods. Our source code is available at: `https://lulusindazc.github.io/econSGproject/`.

## 1 Introduction

The advances in neural 3D scene representation techniques have revolutionized many research and applications in computer vision and graphics. Among these neural 3D scene representation techniques, Neural Radiance Field (NeRF) (Mildenhall et al., 2021) stands out for its ability to learn 3D neural fields directly from 2D images with excellent performance in important real-world applications such as novel view synthesis. Recently, the explicit 3D Gaussian Splatting (3DGS) (Kerbl et al., 2023) has been proposed as an alternative to the implicit NeRF. This technique has demonstrated remarkable reconstruction quality while maintaining high training and rendering efficiency. Concurrent to neural 3D scene representation techniques, large visual-language models (VLMs) such as the CLIP model (Radford et al., 2021) have shown extremely strong capability in zero-shot transfer to the open-world setting for various downstream tasks such as image semantic segmentation, *etc*.

Neural 3D scene representation and multi-modality foundation models are rapidly advancing in parallel. This progress has led to research on open-vocabulary 3D scene understanding, where the neural rendering capabilities of neural fields are leveraged to align VLMs with 3D scenes. To this end, almost all existing works (Kerr et al., 2023; Liu et al., 2024; Qin et al., 2023; Liao et al., 2024; Shi et al., 2023; Zhou et al., 2024; Ye et al., 2023; Guo et al., 2024) unanimously adhered to the fundamental pipeline of first extracting semantic features from the given multi-view images using open-world 2D visual-language models (VLMs), followed by using these semantic features to train semantic fields appended to NeRF or 3DGS. However, since 2D semantic features extracted independently from multi-view images can be incomplete and inconsistent, the primary focus of most existing works is on extracting precise semantic features from VLMs and then efficiently consolidating them into a multi-view consistent 3D neural field representation.

Early approach LeRF (Kerr et al., 2023) leverages CLIP to get semantic features from each of the input multi-view images. However, this often results in semantic features with ambiguous boundaries since CLIP is trained on image-level captions despite the attempt in LeRF to improve granularity with multi-scale CLIP features. Several subsequent works utilize Segment Anything Model (SAM) (Kirillov et al., 2023) or DINOv2 (Oquab et al., 2023) to improve the precision of the semantic features from CLIP. 3DOVS (Liu et al., 2024) and LEGaussians (Shi et al., 2023) use semantic features with better boundaries from DINO to complement CLIP. Feature-3DGS (Zhou et al., 2024) extracts semantic features from either SAM or LSeg (Li et al., 2022). Gaussian Grouping (Ye et al., 2023) leverages only SAM masks, leading to class-agnostic segmentation. Semantic Gaussian (Guo et al., 2024) and OV-NeRF (Liao et al., 2024) unify 2D CLIP features with class-agnostic instance masks generated from SAM. All the above-mentioned works over-trusted DINOv2 or SAM without making any refinement, which we show empirically (*cf.* Fig. 3 Column (b) shows missing regions in the mask proposals from SAM) to be imperfect.

Several approaches such as OV-NeRF (Liao et al., 2024), LeRF (Kerr et al., 2023), 3DOVS (Liu et al., 2024) and Feature-3DGS (Zhou et al., 2024) naively adopt the same dimension for the 3D neural semantic fields as the high-dimensional semantic features from 2D VLMs, which inevitably incurs high computational complexity for training and querying. Methods such as LangSplat (Qin et al., 2023) and LeGaussians (Shi et al., 2023) propose the use of autoencoder or quantization to reduce the dimension of the multi-view 2D semantic features, and therefore result in similar reduction of dimension in the 3D neural semantic fields for efficient computation. However, the reduction of feature dimension is carried out in the 2D space before lifting into the 3D space, and this can lead to multi-view inconsistency that hurts performance. Although Gaussian Grouping (Ye et al., 2023) is efficient by learning 3DGS only for class-agnostic mask rendering, it consequently lacks semantic language information for each Gaussian.

In this paper, we propose **E**fficient and Multi-view **Con**sistent 3D **S**emantic **G**aussians (econSG), a simple yet effective zero-shot model for 3D semantic understanding. Our proposed econSG consists of: 1) **Confidence-region Guided Regularization (CRR)** to alleviate the incompleteness and ambiguous boundaries of the semantic features obtained from VLMs. In contrast to other approaches which over-trusted SAM or DINO, our CRR is designed to get the best from both worlds of OpenSeg (Ghiasi et al., 2022) and SAM with strong 3D multi-view consistency. Specifically, our CRR first backprojects high confidence OpenSeg semantic features from multiple views using the depth maps obtained from Colmap (Schönberger et al., 2016). We then fit a bounding box on the fused features reprojected onto each view to prompt SAM for better region masks. These masks help refine the OpenSeg semantic features towards well-defined boundaries. 2) **Low-Dimensional 3D Contextual Space** to enforce 3D multi-view consistency and enhance computational efficiency. We build a 3D contextual space from 3D features obtained by fusing the backprojected multi-view 2D features instead of operating on each 2D view separately. We then pre-train an autoencoder to get the low-dimensional latent semantic space for initializing the 3DGS semantic fields. The encoder of the pre-trained autoencoder is also used to project CRR-refined semantic features into the same latent space to supervise the 3DGS semantic fields. Our model improves efficiency by enabling strong initialization for 3DGS semantic fields while performing optimization and rendering entirely within the low-dimensional latent space.

We summarize our **main contributions** as follows: 1) We propose a Confidence-region Guided Regularization (CRR) to get 2D semantic features with complete and precise boundaries by mutual guidance from OpenSeg and SAM with strong awareness of multi-view consistency. 2) We design an autoencoder with one-time pre-training to get the low-dimensional 3D contextual space for initialization of the 3D neural semantic fields, and enforce multi-view consistency by backprojecting 2D features from CRR into the same dimension as the low-dimensional 3D contextual space for efficient training. 3) Our econSG show state-of-the-art performance on four benchmark datasets compared to existing methods. Furthermore, we are also the most efficient training among all methods.

## 2 RELATED WORK

**2D Open-vocabulary Segmentation.** 2D open-vocabulary segmentation has seen considerable growth due to the availability of vast text-image datasets and computational resources. Advancements in large VLMs (Alayrac et al., 2022; Jia et al., 2021; Radford et al., 2021) have significantly

enhanced zero-shot 2D scene understanding, even for long-tail objects in images. A common approach for zero-shot predictions is to use vision-and-language cross-modal encoders, which are trained to map images and text labels into a unified semantic space. However, these models often produce embeddings at the image level, which are not suitable for tasks requiring pixel-level information. Recent efforts (Ghiasi et al., 2022; Kuo et al., 2022; Li et al., 2022; Zhou et al., 2022; Rao et al., 2022) aim to bridge this gap by correlating dense image features with language model embeddings, enabling users to detect, classify or segment objects in images with arbitrary text labels. Predominant open-vocabulary segmentation methods (*e.g.* LSeg (Li et al., 2022)) often rely on distilling knowledge from large-scale pre-trained models such as image-text contrastive learning models (*e.g.* CLIP (Radford et al., 2021)) and diffusion models(Rombach et al., 2022). These approaches leverage the rich semantic information captured during pre-training to perform segmentation tasks. However, the distillation process necessitates fine-tuning on specific datasets with a limited vocabulary which undermines the open-vocabulary capability and results in reduced performance in recognizing rare classes. OpenSeg (Ghiasi et al., 2022) utilizes weak supervision through image captions without fine-tuning on a specific class set, but its vocabulary is limited compared to CLIP due to a smaller training dataset. In contrast, our method bypasses fine-tuning CLIP and effectively handles open world classes.

**3D Open-vocabulary Segmentation.** The success of 2D open-vocabulary segmentation has inspired many recent works (Peng et al., 2023; Ding et al., 2023; Nguyen et al., 2024; Takmaz et al., 2023) on 3D open-vocabulary segmentation for point clouds. Many of these methods follow a common design principle: aligning pre-trained 2D open-vocabulary segmentation frameworks such as LSeg (Li et al., 2022) with point cloud feature embeddings. These works primarily rely on point clouds that are relatively more difficult to obtain than multi-view images. To enable multi-view images as input, there has been a significant increase in NeRF-based (Mildenhall et al., 2021) 3D segmentation. Given that the 2D semantic features derived independently from multi-view images are prone to inconsistency, the primary objective is to learn a shared 3D neural representation that enforces consistency by fitting multi-view data into a unified representation with a loss function that penalizes inconsistencies among views. Semantic-NeRF (Zhi et al., 2021) constructs a semantic field which enables the synthesis of segmentation masks from novel views. However, this method requires a large number of annotated labels, which is non-trivial and costed. Some methods (Tschernezki et al., 2022; Fan et al., 2022) utilize the self-supervised feature extractor (*e.g.* DINO (Caron et al., 2021)) to extract 2D features and distill features into the semantic field. More recently, several NeRF-based works (Kerr et al., 2023; Liu et al., 2024; Kobayashi et al., 2022) have explored textual descriptions combined with CLIP models to achieve open-vocabulary 3D semantic understanding. LERF (Kerr et al., 2023) grounds the language field within NeRF by optimizing multi-scale embeddings from CLIP into 3D scenes. 3DOVS (Liu et al., 2024) distills open-vocabulary multimodal knowledge from CLIP and object boundary information from DINO into the NeRF. Wang et al. (2024) focuses on proposing a 3D open-vocabulary segmentation framework that can generalize to unseen scenes. Despite promising results, NeRF-based approaches suffer from slow training and rending. We circumvent these issues by using the more efficient 3D Gaussian Splatting.

**3D Gaussian Splatting.** 3D Gaussian Splatting (3DGS) (Kerbl et al., 2023) has recently gained popularity as a technique for real-time radiance field rendering and 3D scene reconstruction. Inspired by the success of 3DGS in novel view synthesis, various works (Luiten et al., 2023; Yi et al., 2023; Ye et al., 2023) have adapted it for various tasks to leverage its efficient rendering process. However, Gaussian Splatting methods enabling object-level or semantic understanding of the 3D scene are still under-explored yet meaningful. Gaussian Grouping (Ye et al., 2023) extends 3DGS beyond mere scene appearance and geometry modeling with instance level modeling based on class-agnostic SAM masks. Feature3DGS (Zhou et al., 2024) learns high-dimensional semantic fields in 3D Gaussians using multi-view CLIP features, leading to high computational cost. LangSplat (Qin et al., 2023) and LEGaussians (Shi et al., 2023) encode multi-view features from 2D pre-trained VLMs into 3DGS via different feature dimension reduction techniques. However, these approaches suffer from rendering inefficiencies and 3D semantic inconsistencies across training views. FastLGS (Ji et al., 2025) mitigates multiview inconsistencies by using appearance-based correspondences and mask similarities across views, where they force matched masks across the views to take the same semantic field. In contrast, we do not directly apply inconsistent and imprecise semantics from 2D VLMs across views to optimize 3DGS. Instead, we construct a multi-view consistent 3D embedding space based on geometry for modeling the 3DGS semantic fields.

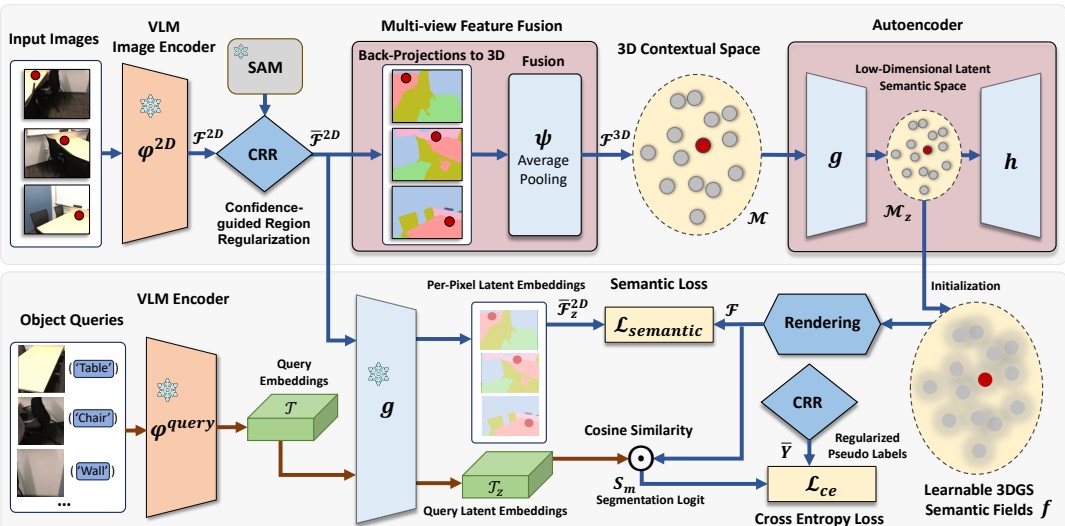

Figure 1: **Our econSG framework.** 1) Top: Building 3D contextual latent space. We use the image encode from a VLM and our CRR to get 2D features $\hat{\mathcal{F}}^{2D}$, which are then back-projected and fused in 3D to get the high dimensional 3D contextual code $\mathcal{M}$. An autoencoder $[g, h]$ is learned to map $\mathcal{M}$ into the low dimensional space $\mathcal{M}_z$. 2) Bottom: 3DGS for semantic fields. We optimize for the 3DGS semantic fields $f$ with $\mathcal{L}_{semantic}$ and $\mathcal{L}_{ce}$ supervised by the image $\hat{\mathcal{F}}^{2D}$ and query $T_z$ latent embeddings obtained by the encoder $g$, respectively. $\mathcal{M}_z$ is used to initialize $f$.

## 3    Preliminaries: 3D Gaussian Splatting

3DGS (Kerbl et al., 2023) explicitly represents the 3D scene as a set of anisotropic 3D Gaussians, which share similarity with point clouds. Each Gaussian is characterized by a center point vector $\mu \in \mathbb{R}^3$ and a covariance matrix $\Sigma \in \mathbb{R}^{3\times 3}$, which influences a 3D point $x$ in the scene following the 3D Gaussian distribution: $G(x) = \frac{1}{(2\pi)^{\frac{3}{2}}|\Sigma|^{\frac{1}{2}}} e^{-\frac{1}{2}(x-\mu)^\top \Sigma^{-1}(x-\mu)}$. To ensure positive semi-definite $\Sigma$ and differential optimization, $\Sigma = RSS^\top R^\top$ is decomposed into two learnable components: a scaling matrix $S \in \mathbb{R}^3$ and a rotation quaternion matrix $R \in \mathbb{R}^4$. Additionally, each Gaussian is parameterized by an opacity value $o \in \mathbb{R}$ and an appearance feature vector defined by $n$ spherical harmonic (SH) coefficients $\mathcal{C} = \{c_i \in \mathbb{R}^3 \mid i = 1, 2, \ldots, d^2\}$, where $d^2$ is the number of coefficients of SH with degree $d$. For rendering, 3D Gaussians are projected onto the image plane of the given view by the $\alpha-$ blending function as follows: $c = \sum_{i=1}^{n} c_i \alpha_i \prod_{j=1}^{i-1}(1 - \alpha_j)$. $c$ is the final color in the rendered image computed by blending $n$ ordered Gaussians that overlap onto the pixel. $c_i \in \mathbb{R}^3$ represent color computed from SH coefficients in the $i^{\text{th}}$ Gaussian. $\alpha_i$ is obtained by multiplying the projected 2D covariance matrix $\Sigma' \in \mathbb{R}^{2\times 2}$ with the learned opacity. $\Sigma' = JW\Sigma W^\top J^\top$ in the camera coordinates is computed using view transform matrix $W$ and the Jacobian matrix $J$ of the affine approximation of the projective transformation.

## 4    Our Method

**Objective.** Given the posed images $\mathcal{I}$ and the corresponding open-vocabulary queries $\mathcal{T}$ from the frozen text encoder of a VLM, the goal is to synthesize semantic masks from novel views rendered by 3DGS parameterized by $\{x, \mu, R, S, c, f\}$, where $f$ is an additional optimizable semantic field we add to 3DGS.

**Overview.** Fig 1 shows an overview of our econSG which consists of: 1) A **pre-training stage** where we first design the ***Confidence-guided Region Regularization (CRR)*** that mutually refines OpenSeg and SAM to get the 2D semantic features. In contrast to (Ye et al., 2023; Liao et al., 2024; Guo et al., 2024; Liu et al., 2024; Shi et al., 2023), our CRR avoids over-trusting DINO or SAM which we empirically show to be imperfect (*cf.* Fig. 3 Column (b)). We then train an autoencoder for the ***low-dimensional 3D contextual space*** to improve the training and query efficiency of the

3DGS semantic fields in the next stage. Unlike (Qin et al., 2023; Shi et al., 2023) which compress semantic features in the 2D space before 3D fusion, we enhance 3D consistency by first fusing the backprojected 2D semantic features to get the 3D contextual space followed by training an autoencoder to get the low-dimensional 3D contextual space. 2) A **training stage** where we initialize the 3DGS semantic fields with the low-dimensional 3D contextual space, and supervise the training of the rendered low-dimensional 3DGS semantic fields efficiently with the CRR semantic features mapped into the same dimension by the frozen encoder in the pre-trained autoencoder. We also utilize the frozen encoder and CRR to align class semantics with the 3DGS semantic fields.

### 4.1 IMAGE AND TEXT EMBEDDINGS

We obtain per-pixel semantic feature $\mathcal{F}^{2D}$ from the RGB images $\mathcal{I}$ with the image encoder of a 2D VLM $\varphi^{2D} : \mathcal{I} \mapsto \mathcal{F}^{2D}$. Similarly, we use the 2D VLM encoder to get the representative open-vocabulary embeddings $\mathcal{T}$ from multi-view object queries $T$ via $\varphi^{query} : T \mapsto \mathcal{T}$. During training, queries $T$ represent object proposals derived from multi-view inputs, whereas during inference, they correspond to text prompts either provided at inference or specified by the user.

### 4.2 CONFIDENCE-GUIDED REGION REGULARIZATION (CRR)

As shown in Fig. 3, the semantic feature map from OpenSeg (Column (a)) and the regional mask proposals from SAM (Column (b)) can be imperfect due to complex background and occlusion, and thus leading to inconsistent and inaccurate semantics across multiple views. We design our CRR for mutual refinement of the per-pixel semantic feature from the 2D VLM and regional mask proposals from SAM as follows:

a: Select pixel embeddings $\mathcal{F}^{2D}$ across all views with confidence higher than threshold $\tau_1$:
  ▷ $\mathcal{R} \leftarrow \text{SelectConfident}(\mathcal{F}^{2D} > \tau_1)$;

b: Back-project semantic features of each pixel in $\mathcal{R}$ into 3D using depthmaps $\mathcal{D}$ from Colmap. Average-pool back-projected $\mathcal{R}$ to get multi-view consistent semantic features:
  ▷ $\bar{\mathcal{F}}^{3D} \leftarrow \text{AvgPool}(\text{BackProject}(\mathcal{R}, \mathcal{D}))$;

c: Obtain semantic label for each 3D point according to its similarity with the query embeddings. On the reprojected points, do majority voting on the semantic labels and average-pooling on the semantic features to get a set of 2D semantic masks and their corresponding features and labels:
  ▷ $\{\mathcal{P}, \bar{\mathcal{F}}^{2D}, \bar{Y}\} \leftarrow \text{Vote-AvgPool}(\text{Project}((\text{SemanticLabel}(\mathcal{T}, \bar{\mathcal{F}}^{3D})))$;

d: Fit bounding boxes to the re-projected $\mathcal{P}$, and use as input prompts to SAM to get better regional mask proposals:
  ▷ $\mathcal{S} \leftarrow \text{PromptSAM}(\text{FitBBox}(\text{Project}(\mathcal{P})))$;

e: Retain $\mathcal{P}$ with confidence higher than threshold $\tau_2$. Assign the semantic label and feature of the high confidence $\mathcal{P}$ to the improved SAM regional mask proposal $\mathcal{S}$ with the highest IoU score:
  ▷ $\{\mathcal{S}, \bar{\mathcal{F}}^{2D}, \bar{Y}\} \leftarrow \text{MaxIoUScore}(\text{SelectConfident}(\mathcal{P} > \tau_2), \mathcal{S})$;

Note that the semantic features $\mathcal{F}^{2D}$, $\bar{\mathcal{F}}^{3D}$ and $\bar{\mathcal{F}}^{2D}$ share the same dimension since $\bar{\mathcal{F}}^{3D}$ is obtained from average pooling of $\mathcal{F}^{2D}$ from multi-view back-projections, and $\bar{\mathcal{F}}^{2D}$ is from the average-pooling of the reprojected $\bar{\mathcal{F}}^{3D}$ in each mask $\mathcal{P}$. Steps (a)-(c) enforces multi-view consistency in the semantic features from OpenSeg. Step (d) uses the multi-view consistent semantic mask to improve regional mask proposals from SAM. Finally, Step (e) uses the improved regional mask proposals from SAM to further refine the multi-view consistent semantic mask.

### 4.3 LOW-DIMENSIONAL 3D CONTEXTUAL SPACE

**Multi-view Feature Fusion.** For each 3D point obtained from Structure-from-Motion (SfM) using Colmap for the initialization of 3DGS, we compute per 3D point feature $f_p^{3D} = \psi(\bar{f}_i^{2D}, \ldots, \bar{f}_{N_p}^{2D})$ from average pooling $\psi$ the multi-view features of the $N_p$ visible corresponding pixels. We build an

initial 3D contextual space by consolidating all point features corresponding to the point cloud from SfM: $\mathcal{M} = \{f_1^{3D}, \ldots, f_p^{3D}\}$.

**Autoencoder.** A naive direct rendering of the feature fields is very time-consuming due to the high dimensionality of the semantic features $\mathcal{F}^{2D}$ since the latent dimensions in 2D foundation models tend to be very large. This problem is further aggravated in complex 3D scenes with a lot of dense points. We thus pre-train an autoencoder to map the high dimensional 3D contextual space $\mathcal{M}$ into a low-dimensional latent space space $\mathcal{M}_z = \{z_1^{3D}, \ldots, z_p^{3D}\}$ to improve efficiency. Specifically, the encoder $z_p^{3D} = g(f_p^{3D})$ maps feature $f_p^{3D}$ with high dimension to a lower dimension latent vector $z_p^{3D}$. The reconstruction is given by $o_p^f = h(g(f_p^{3D}))$, where $h(\cdot)$ is the decoder and $o_p^f$ is the reconstructed 3D semantic feature. The training objective of the autoencoder on the 3D point features $\mathcal{M}$ is as follows:

$$\mathcal{L}_{ae} = \mathcal{L}_{l2}(f_p^{3D}, o_p^f) + \mathcal{L}_{ce}(\hat{y}, \cos < o_p^f, \mathcal{T} >) + \mathcal{L}_{ce}(\hat{y}, \cos < z_p^f, g(\mathcal{T}) >), \quad (1)$$

where $\mathcal{L}_{l2}$ and $\mathcal{L}_{ce}$ denote the $L2$ loss and cross entropy loss, respectively. Using cosine similarity, $\cos < o_p^f, \mathcal{T} >$ outputs the semantic label of the reconstructed semantic feature based on query embeddings and $\cos < z_p^f, g(\mathcal{T}) >$ outputs the semantic label of the encoded low dimension semantic feature. $\hat{y}$ is the pseudo semantic mask generated from the 2D segmentation model.

### 4.4 3DGS Semantic Fields

After obtaining the pre-trained autoencoder $[g(\cdot), h(\cdot)]$, we use the encoder $g(\cdot)$ to map: 1) The initial 3D contextual features to the low-dimensional 3D contextual space $g : \mathcal{M} \mapsto \mathcal{M}_z$; 2) Per-pixel semantic features to per-pixel low-dimensional semantic features $g : \bar{\mathcal{F}}^{2D} \mapsto \bar{\mathcal{F}}_z^{2D}$ ; 3) Query semantic features to low-dimensional query semantic features $g : \mathcal{T} \mapsto \mathcal{T}_z$.

We use the low-dimension 3D contextual space $\mathcal{M}_z$ to initialize the semantic field $f$ in each 3D Gaussians, and render the 3DGS semantic fields into each view via alpha-blending:

$$\mathcal{F} = \sum_{i \in n} f_i \alpha_i \prod_{j=1}^{i-1} (1 - \alpha_j). \quad (2)$$

We supervise the rendered semantic fields $\mathcal{F}$ by their semantic logit $S_m$ with the semantic mask label $\bar{Y}$ from CRR using a cross-entropy loss: $\mathcal{L}_{ce} = \mathrm{CE}(S_m, \bar{Y})$, where the semantic logit is obtained from the cosine similarity between the low-dimensional semantic and query features: $S_m = \cos < \mathcal{F}, \mathcal{T}_z >$. Furthermore, we optionally regularize $\mathcal{F}$ to improve feature smoothness with the low-dimensional semantic features $\bar{\mathcal{F}}_z^{2D}$ using a L2 semantic loss: $\mathcal{L}_{semantic} = \mathrm{L2}(\mathcal{F}, \bar{\mathcal{F}}_z^{2D})$.

The final supervision loss for optimizing the given scene is formulated as follows:

$$\mathcal{L} = \mathcal{L}_{color} + \lambda_{2d} \mathcal{L}_{ce} + \lambda_{sem} \mathcal{L}_{semantic}, \quad (3)$$

where $\mathcal{L}_{color}$ is the 3D Gaussian image rendering loss, and $\lambda_{2d}$ and $\lambda_{sem}$ denote hyperparameters to balance the loss terms. In inference, we use Eq. 2 to render the learned 3DGS semantic fields from 3D to 2D. We deploy the encoder $g(\cdot)$ from the pre-trained autoencoder to get query features $\mathcal{T}_z$ of open-world queries. By computing activation scores between the rendered 3DGS semantic fields $\mathcal{F}$ and query features, we can obtain open-world segmentation predictions.

## 5 Experiments

We perform a series of experiments to demonstrate the effectiveness of our proposed method across various 3D scene understanding tasks. We evaluate our method on the 2D semantic segmentation benchmarks: ScanNet (Dai et al., 2017) and Replica (Straub et al., 2019), and 3D open-vocabulary segmentation benchmarks: LERF (Kerr et al., 2023) and 3DOVS (Liu et al., 2024) to compare with previous work, and provide results from ablation studies. We further showcase qualitative results on the Mip-Nerf360 (Barron et al., 2022) for exciting open-vocabulary applications such as 3D object localization, 3D object removal, 3D object inpainting, and language-guided editing.

Table 1: Comparisons of open-vocabulary segmentation on 3DOVS dataset. Best results in **bold**.

| | Dataset | 3DOVS | | | | | | | | | | | |
|---|---|---|---|---|---|---|---|---|---|---|---|---|---|
| | Method | bed | | sofa | | lawn | | room | | bench | | overall | |
| | | mIoU | mAcc | mIoU | mAcc | mIoU | mAcc | mIoU | mAcc | mIoU | mAcc | mIoU | mAcc |
| 2D | LSeg | 56.0 | 87.6 | 4.5 | 16.5 | 17.5 | 77.5 | 19.2 | 46.1 | 6.0 | 42.7 | 20.6 | 54.1 |
| 3D | LERF | 73.5 | 86.9 | 27.0 | 43.8 | 73.7 | 93.5 | 46.6 | 79.8 | 53.2 | 79.7 | 54.8 | 76.7 |
| | 3DOVS | 89.5 | 96.7 | 74.0 | 91.6 | 88.2 | 97.3 | 92.8 | 98.9 | 89.3 | 96.3 | 86.8 | 96.2 |
| | Feature3DGS | 56.6 | 87.5 | 6.7 | 12.4 | 37.3 | 82.6 | 20.5 | 36.7 | 6.2 | 43.0 | 25.5 | 52.4 |
| | LEGaussians | 45.7 | - | 48.2 | - | 49.7 | - | 44.7 | - | 47.4 | - | 47.1 | - |
| | LangSplat | 73.5 | 89.7 | 82.3 | **98.7** | 89.9 | 95.6 | 95.0 | **99.4** | 70.6 | 92.6 | 82.3 | 95.2 |
| | econSG (Ours) | **94.9** | **97.4** | **91.6** | **98.7** | **96.3** | **98.5** | **95.8** | **99.4** | **93.0** | **97.6** | **94.3** | **98.3** |

Table 2: Comparisons of localization accuracy on LERF dataset. Best results in **bold**.

| | Dataset | LERF | | | | |
|---|---|---|---|---|---|---|
| | Method | ramen | figurines | teatime | waldo_kitchen | overall |
| 2D | LSeg | 14.1 | 8.9 | 33.9 | 27.3 | 21.1 |
| 3D | LERF | 62.0 | 75.0 | 84.8 | 72.7 | 73.6 |
| | LangSplat | 73.2 | 80.4 | 88.1 | 95.5 | 84.3 |
| | SemanticGaussian | 76.8 | 83.1 | 89.8 | 90.9 | 85.2 |
| | LEGaussians | 78.6 | 73.7 | 85.6 | 90.1 | 82.0 |
| | econSG (Ours) | **83.2** | **89.3** | **93.4** | **96.2** | **90.5** |

## 5.1 DATASETS AND EXPERIMENTAL SETTING

**Datasets.** To measure the semantic segmentation performance in open-world scenes, we evaluate the effectiveness of our approach using two established multi-view indoor scene datasets: Replica (Straub et al., 2019) and Scannet (Dai et al., 2017), and two 3D open-vocabulary segmentation datasets: LERF (Kerr et al., 2023) and 3DOVS (Liu et al., 2024). For both ScanNet and Replica, we construct training and test sets by evenly sampling sequences in each scene. Images are rendered at the resolution of $640 \times 480$. We adopt 20 different semantic class categories for Scannet by following Openscene (Peng et al., 2023), while Replica is annotated with 51 classes for evaluation as in (Engelmann et al., 2024). For LERF and 3DOVS, we follow the settings in LangSplat (Qin et al., 2023) where LERF is extended with ground truth masks annotated for language queries and 3DOVS consists of $20 \sim 30$ images for each scene with the resolution of $4032 \times 3024$. To assess 3D reconstruction quality, we applied our method to Mip-Nerf360 (Barron et al., 2022) and LERF-Localization (Kerr et al., 2023) by following Gaussian Grouping (Ye et al., 2023).

**Implementation details.** For 2D VLMs, we utilize pixel-level encoders, OpenSeg (Ghiasi et al., 2022) and LSeg(Li et al., 2022) to extract the per-pixel semantic features of each image in indoor scene datasets, and adopt Openclip (Ilharco et al., 2021) to extract image-level features for language-guided editing on Mip-Nerf360, LERF and 3DOVS datasets. We then use SAM for mutual refinement with the 2D VLMs in our CRR to get the semantic features where we set $\tau_1 = 0.45, \tau_2 = 0.6$. We use the Adam optimizer with the learning rate 0.0025 for latent semantic fields. For parameters to train the image scene, we follow the default setting in the original 3DGS (Kerbl et al., 2023). For additional parameters introduced to train the semantic scene, we set $\lambda_{sem} = 1, \lambda_{2d} = 1$. For all datasets, we train each scene for 30K iterations on one NVIDIA RTX-4090 GPU.

## 5.2 MULTIVIEW RECONSTRUCTION AND SEGMENTATION

**Open-vocabulary Segmentation Comparison.** Tab. 1 and Tab. 2 show quantitative results of open-vocabulary segmentation on 3DOVS dataset and localization accuracy on LERF dataset. Tab. 3 shows segmentation comparison across various scenes on both Scannet and Replica. We compare with 2D-based open-vocabulary segmentation model LSeg (Li et al., 2022) along with 3D NeRF and 3DGS-based methods. LERF (Kerr et al., 2023) and 3DOVS (Liu et al., 2024) leverage the multi-scale CLIP features from the image patches as supervisions for learning NeRF-based semantic field, and thus struggle with both object boundary ambiguities for segmentation and rendering efficiency. Their performance on novel views can be greatly degraded because of they generate inconsistent and imprecise ground truth semantics from multi-scale features across multiple views. Feature3DGS (Zhou et al., 2024) directly applies inconsistent and noisy semantics from training

Table 3: Comparison with other methods on segmentation of novel views from Scannet and Replica. Best results highlighted in **bold**.

| Dataset | FPS | Replica | | | | Scannet | | | |
|---|---|---|---|---|---|---|---|---|---|
| | | sparse-view | | multi-view | | sparse-view | | multi-view | |
| | | mIoU | mAcc | mIoU | mAcc | mIoU | mAcc | mIoU | mAcc |
| LERF | 0.2 | 4.312 | 17.080 | 8.285 | 22.125 | 14.059 | 38.734 | 15.349 | 40.294 |
| 3DOVS | 0.3 | 4.553 | 19.356 | 9.081 | 23.938 | 14.227 | 40.584 | 17.802 | 42.532 |
| Feature3DGS | 2.5 | 9.584 | 38.245 | 10.634 | 36.520 | 17.552 | 48.686 | 18.069 | 54.101 |
| econSG (Ours) | 156 | **25.513** | **70.716** | **33.869** | **78.564** | **39.018** | **74.805** | **48.205** | **86.178** |

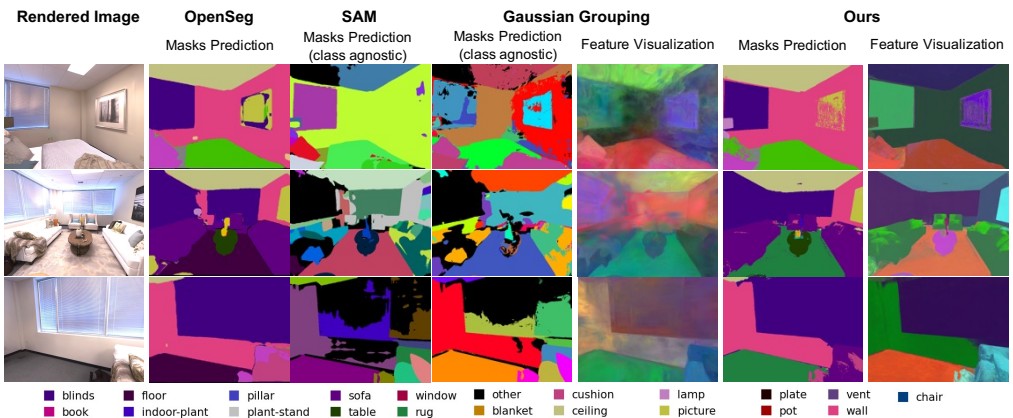

Figure 2: Qualitative comparison of our econSG with Gaussian Grouping (Ye et al., 2023) on Replica.

views to optimize high-dimensional semantic field in 3D Gaussians, resulting in high computation costs and inferior segmentation results. LangSplat (Qin et al., 2023) and LEGaussians (Shi et al., 2023) compress 2D features across all views to improve rendering efficiency on semantic fields, but their performance are still hindered by the inherent 3D semantic noises and inconsistency. SemanticGaussian distills noisy 2D features into an additional 3D model for learning 3D semantics while ignoring semantic consistency from the multi-view 2D images. Our model consistently shows the best performance since we introduce the 3D contextual latent space to provide sufficient 3D semantic consistency into the ground truths and design a CRR step to generate clean and complete semantic masks. These components help ensure optimization efficiency and robustness even with few input images.

In Tab. 3, we also present the inference speed under the multi-view setting in terms of the frames per second (FPS) metric. NeRF-based methods are generally constrained on rendering efficiency and slow. 3DGS-based models are inefficient from high-dimensional language features in 3D Gaussians. We also perform robustness comparison by evenly sampling sparse training views for optimization(30 images per-scene in our experiments). It shows our model consistently outperforms other methods, proving the proposed components help ensure optimization efficiency and robustness even with few input images. In Fig. 2, we visualize the learned semantic fields by showing the rendered latent embeddings in the testing views. We observe that our predictions are of better consistency across views with more complete and well-defined boundaries semantics masks.

**Ablation on CRR.** We compare our CRR with OpengSeg and SAM, and conduct ablation studies on CRR. OpenSeg in Fig. 3(a) shows issues such as ambiguous boundaries and inaccurate dense predictions. This is due to the use of noisy segmentation maps from pre-trained visual encoders for supervision. Fig. 3(b) shows that a naive over-trusting of SAM masks to refine boundaries does not work well in complex scenes. Fig. 3(c) *vs.* (d) and Fig. 3(e) *vs.* (f) show the without and with our CRR on the training and testing sets, respectively. We can see that our CRR effectively produces semantic fields with clear boundaries.

**Analysis on 3D Contextual Latent Space.** We show qualitative results of 3D segmentation predictions and contextual feature space in Fig. 4. The 3D segmentations derived from the original

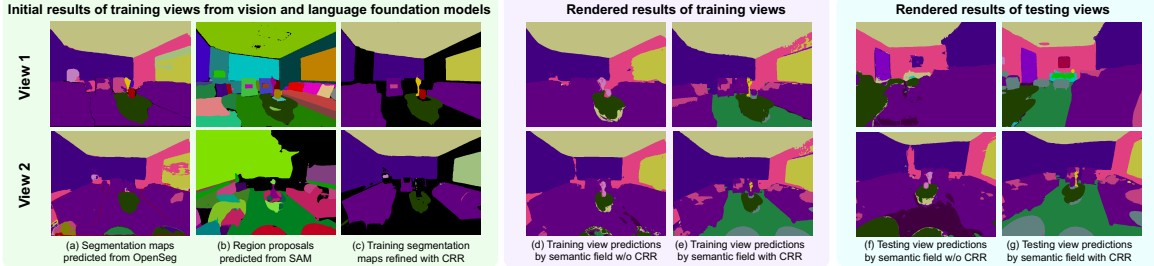

Figure 3: Ablation on confidence-guided region regularization (CRR) with qualitative results of our econSG on Replica. Panels (a)-(e) are from training views, and panels (f)-(g) are from testing views.

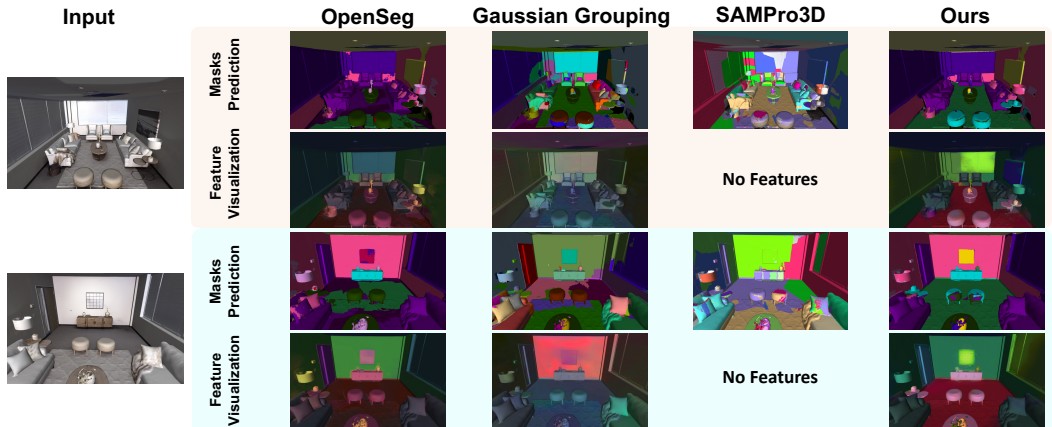

Figure 4: Qualitative 3D Segmentation results and comparison of our method. The second and fourth rows illustrate the feature visualization in 3D space.

OpenSeg exhibit significant coarseness and errors due to multi-view inconsistency among the predicted 2D semantic features. Gaussian Grouping shows better object-level boundaries by leveraging SAM object mask IDs as direct supervision. However, SAM can fail in complex scenes leading to incorrect masks in some views. Morever, since SAM segmentations are class-agnostic, the learned 3D semantic embeddings from Gaussian Grouping are only instance-level and cannot be queried by text embeddings. SAMPro3D (Xu et al., 2023) proposes to filter low-quality prompts and consolidate prompts inside the object. However, SAMPro3D is not applicable to open-vocabulary 3D scene understanding tasks without feature embeddings. In contrast, our model significantly improves the quality of 3D contextual space and segmentation predictions as illustrated in the last column.

**Training Efficiency Analysis.** In Tab. 4 , we show training and inference time on the "sofa scene" of 3DOVS dataset at different feature dimensions. Compared with LangSplat, our model achieves a significant speed increase in inference (LangSplat:401.9s *vs.* Ours:4.9s). This is because LangSplat performs evaluation on the original high-dimensional space while our model directly makes predictions in the low-dimensional latent contextual space. Our model can achieve promising efficiency and accuracy due to the low-dimensional 3D latent contextual space that avoids the need for training high-dimensional 3DGS semantic fields. The last column of Tab 4 shows that the high-dimensional features (*e.g.* 512 for CLIP features) pose huge memory and computation demands especially on training when the autoencoder is removed.

## 5.3 APPLICATIONS

**3D Scene Editing.** Fig. 5 (Right) shows examples of language-guided editing on a object scene from the Bear data (Ye et al., 2023) and a room scene from Mip-NeRF360 (Barron et al., 2022). We utilize the text encoder to embed the object category names to identify the corresponding 3DGS points and adjust their attributes such as coordinates and colors. We first detect regions that are invisible in all

Table 4: Training efficiency analysis on the sofa scene of the 3DOVS dataset.

| Methods | LERF | 3DOVS | Langsplat | Feature3DGS | Ours | | | Ours (remove autoencoder) |
|---|---|---|---|---|---|---|---|---|
| Feature dimension | 512 | 512 | 3 | 128 | 6 | 16 | 32 | 512 |
| mIoU (%) | 27.0 | 74.0 | 82.3 | 6.7 | 91.6 | 91.8 | 91.8 | OOM |
| Training time (min) | 19.4 | 78 | 66 | 87 | 29 | 32 | 43 | OOM |
| Inference (s) | 121.4 | 6.6 | 401.9 | 6.0 | 4.9 | 5.2 | 5.3 | OOM |

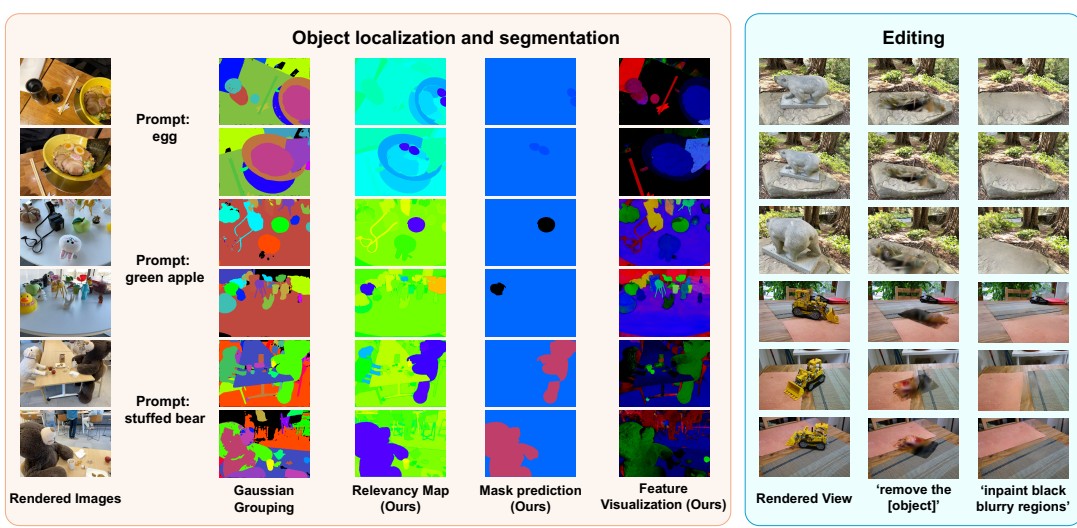

Figure 5: Qualitative examples of language-guided segmentation and editing. Segmentation results of the rendering views are compared with Gaussian Grouping on LERF-localization dataset.

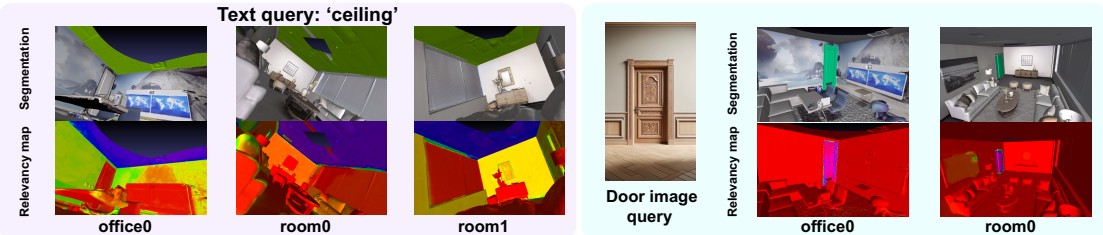

Figure 6: A 3D scene can be queried using text prompt embedding or images to locate matching 3D points. Colors of image query outlines indicate corresponding matches in the 3D point cloud.

views after deletion and then inpaint these specific areas instead of the entire 2D object regions. We then use the 2D inpainted image in each rendering view to guide the learning of new 3D Gaussians.

**Open-Vocabulary 3D Object detection.** Fig. 5 (Left) shows examples of object localization and segmentation with text queries. In Fig. 6, we query a 3D scene database to retrieve examples based on their similarity to a given input image. We first encode the query text or image using CLIP image encoder and then threshold the cosine similarities between the CLIP features and the 3DGS semantic fields to produce a 3D object detection and mask.

## 6 CONCLUSION

In this paper, we propose econSG for open-vocabulary semantic segmentation of 3D scenes. Specifically, we propose CRR to get 2D semantic features with complete and precise boundaries by mutual guidance from OpenSeg and SAM with strong awareness of multi-view consistency. We design an autoencoder with one-time pretraining to get the low-dimensional 3D contextual space for initialization of the 3D neural semantic fields, and enforce multi-view consistency by backprojecting 2D features from CRR into the same dimension as the low-dimensional 3D contextual space for efficient training. Our econSG shows state-of-the-art performance on four benchmark datasets compared to the existing methods. Furthermore, we are also the most efficient training among all the methods.

**Acknowledgments.**  This research work is supported by the Agency for Science, Technology and Research (A*STAR) under its MTC Programmatic Funds (Grant No. M23L7b0021), and the Tier 2 grant MOE-T2EP20124-0015 from the Singapore Ministry of Education.

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

# A APPENDIX

## A.1 IMPLEMENTATION

Our auto-encoder is implemented by MLPs, which compresses the high-dimensional CLIP features into low-dimensional latent features, *i.e.* 738 for OpenSeg and 512 for CLIP features. The encoder consist of layers with size as follows: [256, 128, 64, 32, 6]. The decoder is composed with layers by the following dimensions: [16, 32, 64, 128, 256, 256, 768].

## A.2 ABLATION STUDY ON CRR.

We perform the ablation study of CRR on Scannet and Replica in Tab. 5. In row 'w.CRR(w.o Step a.)', we evaluate Step a of CRR which performs high-confidence region selection for computing 3D semantic features. In row 'w.CRR (w.o Step d.)', we analyze Step d of CRR which further use SAM to refine the projected 2D segmentation maps. For example, in Room0 and scene0494 scenes, our CRR improves performance greatly in terms of both mIoU, demonstrating that CRR helps refine segmentation boundaries.

Table 5: Ablation study for CRR on Scannet and Replica.

| Setting | Room0 | | Replica | | scene0494 | | Scannet | |
|---|---|---|---|---|---|---|---|---|
| | mIoU | mACC | mIoU | mACC | mIoU | mACC | mIoU | mACC |
| w.o CRR | 13.245 | 41.307 | 10.604 | 29.816 | 18.674 | 36.869 | 17.933 | 42.133 |
| w. CRR (w.o step a.) | 22.336 | 61.563 | 23.276 | 69.364 | 52.861 | 84.450 | 36.549 | 75.235 |
| w. CRR (w.o step d.) | 27.091 | 68.647 | 27.146 | 72.843 | 57.924 | 86.583 | 42.098 | 83.476 |
| w CRR | 31.715 | 72.492 | 33.869 | 78.564 | 63.043 | 90.574 | 48.205 | 86.178 |

## A.3 MORE VISUALIZATION RESULTS

We demonstrate more examples on Scannet dataset for open-vocabulary 3D semantic segmentation in Fig. 7. In Fig. 8, Fig. 9 and Fig. 10, we present additional examples of retrieved objects from the 3DOVS and LERF datasets. Our model consistently provides clearer semantic patterns and exhibits reduced noise compared to LEGaussians and LangSplat. These results demonstrate the effectiveness of our model.

## A.4 3D MULTIVIEW CONSISTENCY OF OUR CRR

Fig. 11 shows an illustration of a 3D point with semantic label that is consistent (Top) and inconsistent (Bottom) across all multiple views. The steps in our CRR modify the results from OpenSeg/LSeg and SAM to ensure that the 2D features are 3D consistent across all multiple views (Top case). In contrast, other baseline methods overtrusted SAM and/or OpenSeg/LSeg often result in 3D multiview inconsistencies (Bottom case). Consequently, the supervision of 3DGS semantic fields from the features of our CRR tend to lead to good performances.

## A.5 VISUALIZATION RESULTS

In Fig.12, we present the open-vocabulary segmentation results on the 3DOVS(Liu et al., 2024) and LERF (Kerr et al., 2023) datasets using object attributes such as color, texture, and function as queries. For example, when queried with the attribute "plush" in the LERF-teatime scene, our model successfully localizes both the plush sheep and plush bear with accurate segmentation boundaries. When queried with "white" on the same scene, our model correctly localizes the white plush sheep. This demonstrates the robustness of our model in handling diverse open-vocabulary text queries.

In Fig. 13, we show the visualization results on MVImgNet (Yu et al., 2023) datasets for open-vocabulary segmentation using part-level text queries. We chose MVImgNet to show this task because it contains single object scenes. Open-vocabulary segmentation using part-level text queries is too challenging on the LeRF and 3DOVS datasets which contain multiple objects in the scenes. In our experiments and as shown in the figures, we use object part names as text queries to segment

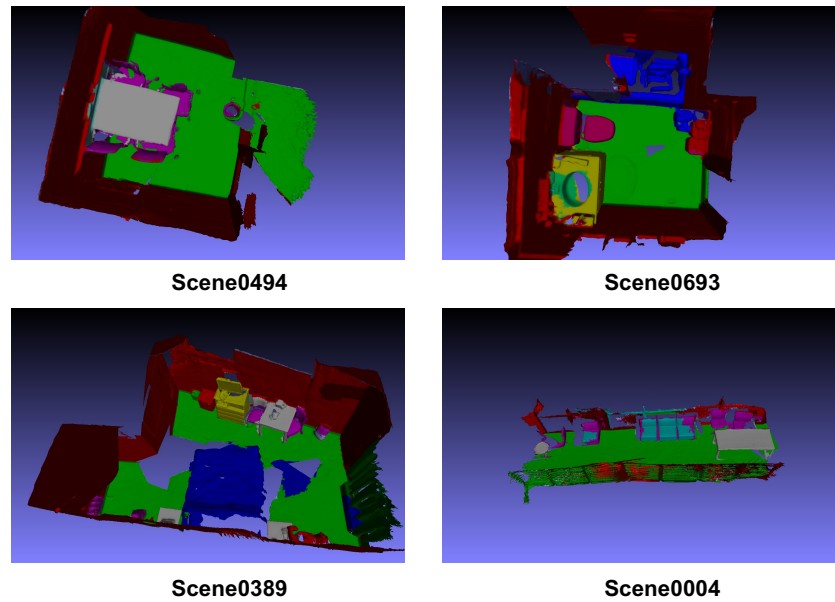

Figure 7: Qualitative 3D segmentation results of our econSG on the Scannet dataset.

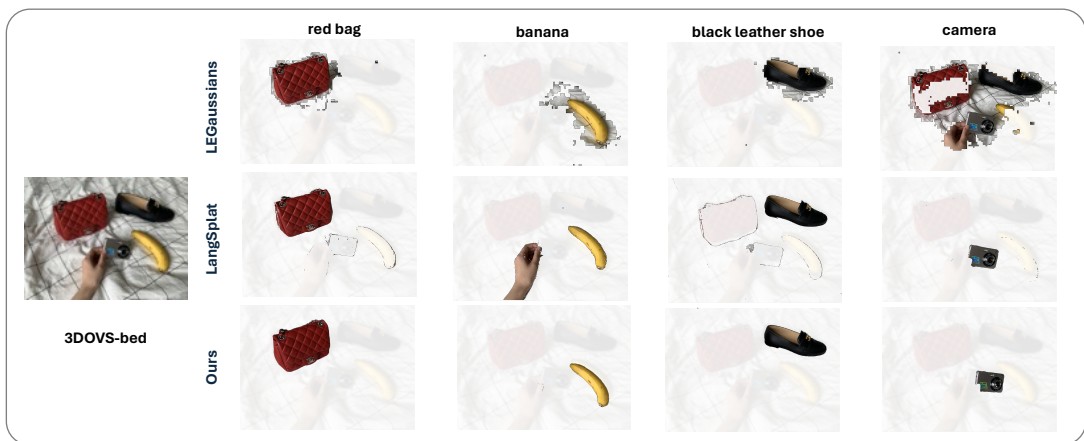

Figure 8: Qualitative comparison of our econSG with baselines on the 3DOVS dataset. We show the visualization of the retrieved objects in the scene. The quantitative results are in Table 1 in the main paper.

the object parts for evaluation. LangSplat generates three scales with SAM and performs evaluations using 3DGS models at small, medium, and large scales during inference to select the optimal scale for a query. The results show that LangSplat struggles with open-vocabulary part segmentation. For example, when queried with "guitar headstock" (third row), LangSplat identifies the entire guitar instead of the headstock. In contrast, our econSG generates precise segmentation maps for each part-level query. Although our econSG is capable of showing precise part segmentation for this example, we do not claim the full capability of part segmentation. This challenging task is out-of-scope and we will leave it for future work.

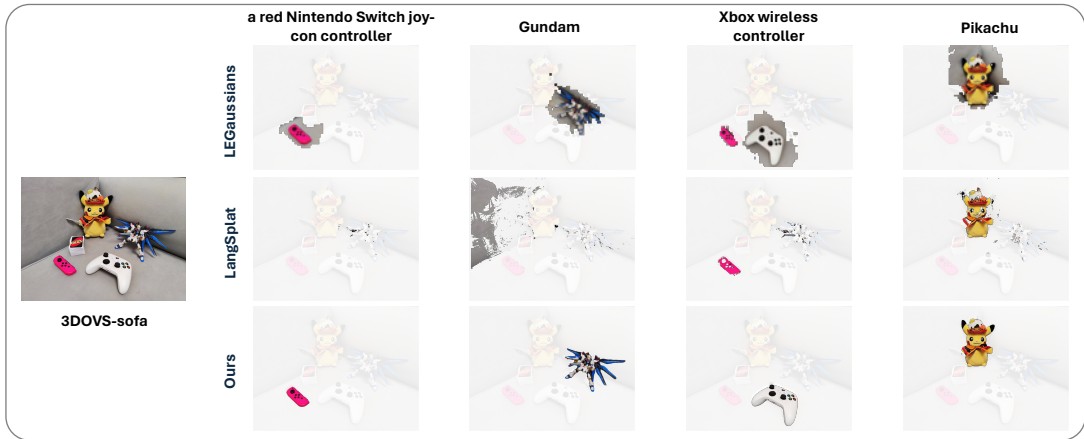

Figure 9: Qualitative comparison of our econSG with baselines on the 3DOVS dataset. We show the visualization of the retrieved objects in the scene. The quantitative results are in Table 1 in the main paper.

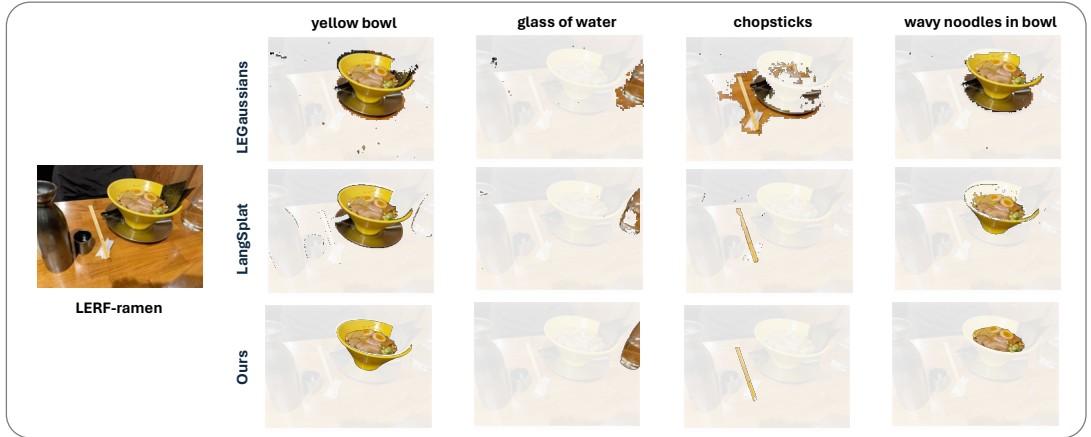

Figure 10: Qualitative comparison of our econSG with baselines on the LERF dataset. We show the visualization of the retrieved objects in the scene. The quantitative results are in Table 2 in the main paper.

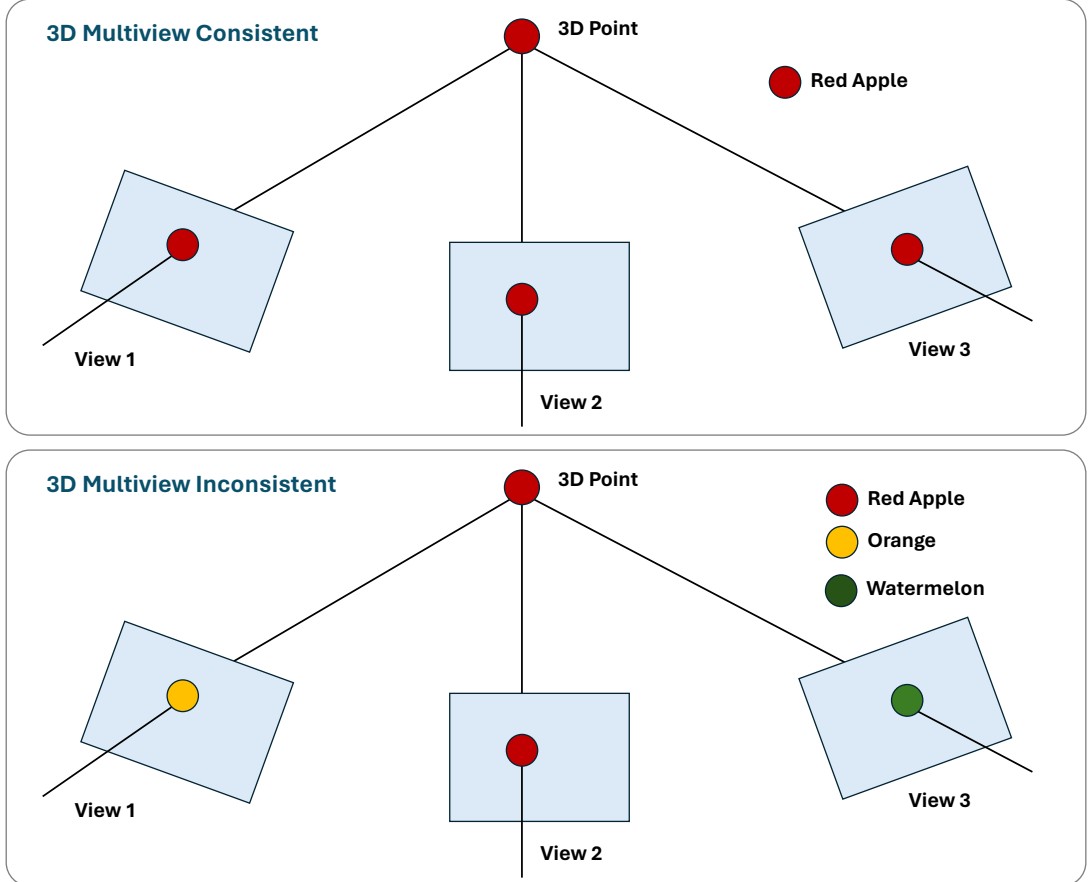

Figure 11: **Multiview consistency illustration.** Top: The semantic label of a 3D point matches the semantic labels in the 2D images across multiple views. Bottom: The semantic label of the 3D point do not match all the 2D semantic labels in the 2D images across multiple views.

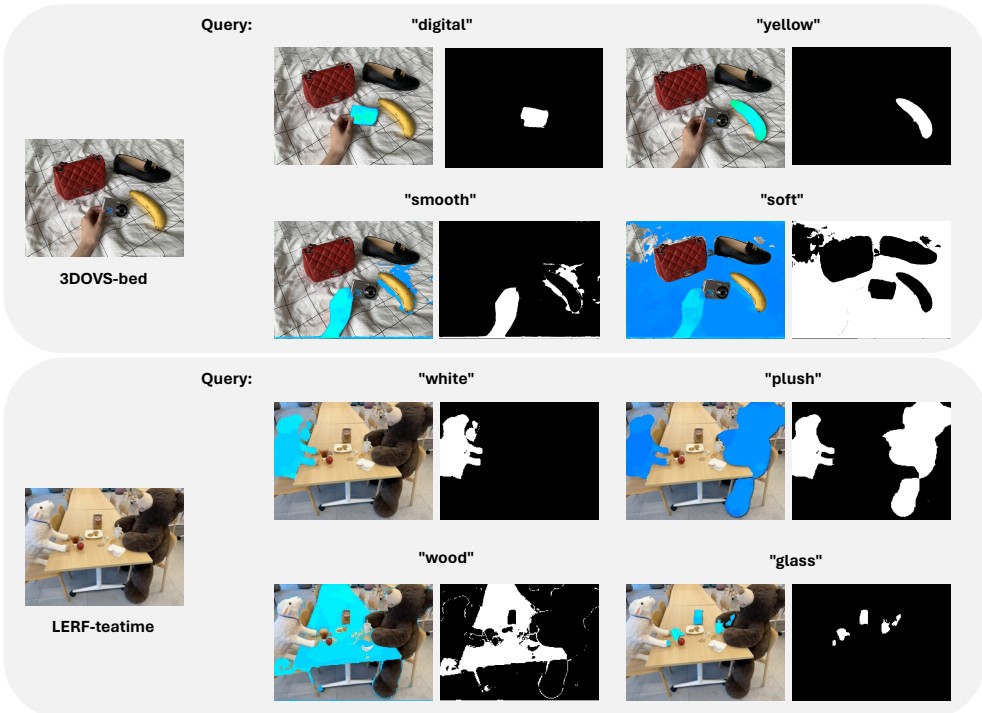

Figure 12: Qualitative results (relevancy maps and segmentations) of our econSG for evaluating the open-vocabulary attribute query.

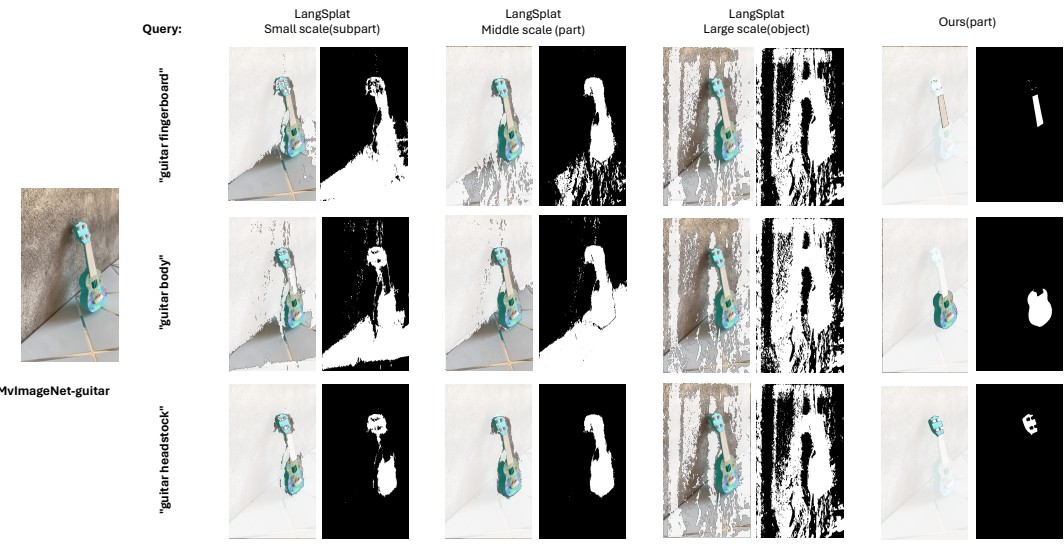

Figure 13: Qualitative comparisons between our econSG and LangSplat on MVImgNet dataset for evaluating the open-vocabulary 3D part segmentation.

