# OpenReview forum: "econSG: Efficient and Multi-view Consistent Open-Vocabulary 3D Semantic Gaussians"
_ICLR.cc/2025/Conference — ICLR 2025 Poster_

### Official Review · Reviewer_ZWwm · 2024-10-27

**Soundness:** 3
**Presentation:** 2
**Contribution:** 3
**Rating:** 6
**Confidence:** 3

**Summary:**

This paper introduces econSG, a novel 3D semantic Gaussian model that tackles the challenges of extracting precise semantic features and maintaining multi-view consistency in open-vocabulary neural fields. The proposed method incorporates Confidence-region Guided Regularization (CRR) and a low-dimensional contextual space, resulting in state-of-the-art performance on four benchmark datasets. Notably, econSG also exhibits superior efficiency in training and Inference compared to existing methods.

**Strengths:**

+ The paper introduces CRR as a novel approach to refine semantic features from VLMs, specifically focusing on achieving precise boundaries and completeness. The mutual refinement strategy between OpenSeg and SAM is a significant innovation that addresses the limitations of existing methods in capturing accurate and complete semantic information.
+ By constructing a low-dimensional 3D contextual space, econSG effectively integrates 2D features from different views, improving the consistency and accuracy of 3D scene representation.
+ EconSG significantly reduces computational complexity and improves training and querying efficiency by mapping high-dimensional features to a low-dimensional space using a pre-trained autoencoder.
+ The paper demonstrates a notable  improvement in performance, achieving state-of-the-art results on four benchmark datasets.

**Weaknesses:**

- Despite mentioning an ablation study on CRR, the paper does not include the corresponding experimental outcomes. This gap is detrimental to evaluating the specific impact of each CRR component on the overall effectiveness of the method.
- Figure 2 shows that the method proposed in this article tends to segment large objects and directly ignores small objects?

**Questions:**

Since the article repeatedly emphasizes the multi-view consistency, I suggest adding some introductions to related research in the related work or introduction.

---

### Official Review · Reviewer_C9FH · 2024-10-29

**Soundness:** 2
**Presentation:** 2
**Contribution:** 2
**Rating:** 6
**Confidence:** 3

**Summary:**

This paper aims to refine visual foundation models' output and solve the multi-view inconsistency problem in 3D rendering. Specifically, they proposed a confidence-region guided regularization and back projected 2D features into a latent space shared with 3D features. Rendering is performed via alpha blending, supervised by semantic information, 2D features, and color information.

**Strengths:**

1. Experiment results seem to be good.
2. Incorporating visual foundation models into this field for multi-modal learning is interesting.

**Weaknesses:**

Currently, I vote for 5, marginally below the acceptance threshold, for the reasons below:

1. My primary concern is the lack of explanation on the open-vocabulary experiment settings. In section 5.1, this manuscript does not clarify how their sampling method ensures an open-vocabulary setting or specify which classes are used in training. It will be helpful to provide more explanation on experiments. Additionally, in Table 1, only a few classes are listed for the class-level comparisons. From my current knowledge, they adopted ‘20 different semantic class categories’ in Scannet. It will be beneficial to include the open-vocabulary experiment results in more classes.

2. Writing. Some parts of the manuscript will benefit from fine-tuning for brevity, i.e., ‘The parallel rapid developments of neural 3D scene representation and large multi-modality foundation models naturally lead to research on open-vocabulary 3D scene understanding by leveraging the neural rendering capability of neural fields to align the visual-language models to 3D scenes’. Condensing long sentences can better retain reader’s attention on the main point. Another small note: it seems that there is a typo in section 4.1 in the text feature’s notation T.

3. Back projecting 2D information into a shared latent space with 3D is not entirely novel. It would be helpful to provide more explanation on their unique contribution for low-dimensional 3D contextual space.

**Questions:**

Please refer to the weakness part.

---

### Official Review · Reviewer_Mv8f · 2024-10-30

**Soundness:** 2
**Presentation:** 3
**Contribution:** 2
**Rating:** 6
**Confidence:** 4

**Summary:**

This paper introduces econSG, a new approach for open-vocabulary semantic segmentation within 3D Gaussian Splatting (3DGS), addressing challenges with consistency and efficiency in recent methods. Most current open-vocabulary neural fields rely heavily on Segment Anything Model (SAM) to regularize image-level features extracted from CLIP, often without additional refinement, which can lead to inconsistencies across multiple views. Additionally, some methods use dimensionality reduction on 2D semantic features before fusing them with 3D semantic fields, which compromises multi-view consistency.

EconSG overcomes these issues through two main innovations:

1. Confidence-region Guided Regularization (CRR): This approach mutually refines SAM and CLIP features to achieve complete and precise semantic boundaries with enhanced accuracy.
2. Low-dimensional Contextual Space: By fusing multi-view 2D features into a single 3D representation before dimensionality reduction, econSG enhances multi-view consistency and computational efficiency.

**Strengths:**

This paper clearly explains its motivation and methodology, with both the algorithmic process and pipeline illustrations presented in a way that is very clear to the reviewers.

**Weaknesses:**

1. I believe the second contribution Low-dimensional Contextual Space lacks novelty. Similar idea of mapping high-dimensional features into a low-dimensional space in this task can be seen in previous work like one of the baselines, LangSplat[1]. Even if the authors claim that the contribution is to increase the inference speed by changing the space of rendering, recent work like FastLGS: Speeding up Language Embedded Gaussians with Feature Grid Mapping[2] also proposes the same idea of directly rendering low dimensional semantic features.
2. The qualitative and quantitative experiments are not well-aligned. The baselines that appeared in the quantitative tables are not mentioned in the qualitative comparisons, which reduces the reliability of data in the tables. If the author can provide qualitative results that can align with the quantitative results and more qualitative results that display your 3D consistency, this work can be more convincing.

[1] Qin, M., Li, W., Zhou, J., Wang, H., & Pfister, H. (2024). Langsplat: 3d language gaussian splatting. In Proceedings of the IEEE/CVF Conference on Computer Vision and Pattern Recognition (pp. 20051-20060).
[2] Ji, Yuzhou, et al. "FastLGS: Speeding up Language Embedded Gaussians with Feature Grid Mapping." arXiv preprint arXiv:2406.01916 (2024).

**Questions:**

I'm confused with the 3D consistency brought by the CRR module.
- For step b. From my understanding, the consistency here highly depends on the accuracy of the COLMAP depth maps. I understand that average pooling can partially mitigate issues like complex backgrounds or occlusion in individual views, resulting in more stable, 3D-consistent semantic features. But if some view features are affected by occlusion, noise, or low confidence, simple averaging may dilute the features, potentially affecting consistency. Moreover, if the features from different views have large discrepancies, average pooling alone may not fully eliminate these differences.
- For step c. If a reprojected point is occluded in most of the views, I believe your majority voting strategy may assign a label of the occlusion  to that point instead of the correct label of that point.
Can you resolve my confusion about the two points mentioned above? Please elaborate on how you ensure your 3D consistency and how your major voting strategy is implemented. Thanks.

**Details Of Ethics Concerns:**

No ethics review is needed.

---

### Official Review · Reviewer_9XDx · 2024-11-01

**Soundness:** 1
**Presentation:** 2
**Contribution:** 2
**Rating:** 3
**Confidence:** 3

**Summary:**

This paper presents a novel approach for open-vocabulary neural field segmentation. Current methods face two key challenges: 1) they rely too heavily on the inconsistent 2D features from 2D VLMs, and 2) they are computationally expensive due to the need for high-dimensional feature rendering. To address these, the authors propose a Confidence-region Guided Regularization technique to achieve precise boundaries and multi-view consistency, as well as a 3D autoencoder to project high-dimensional features into a lower-dimensional space. The proposed method is evaluated against the current state-of-the-art across multiple benchmarks.

**Strengths:**

1. The paper is well-motivated, addressing the issues of 2D feature inconsistency and the high computational cost associated with high-dimensional features.
2. The paper provides extensive benchmarking against current state-of-the-art methods.

**Weaknesses:**

1. The implementation details of the proposed 3D autoencoder are unclear. Could you clarify the architecture of the 3D autoencoder (e.g., is it an MLP or another type of model), and specify its number of parameters? Additionally, how is the autoencoder trained—is it trained per scene? What is the duration of the training process?
2. During inference, text features are also projected into a low-dimensional space for querying. However, since the autoencoder is not trained on a rich set of diverse text features, information loss is likely during encoding, which may impair the model's ability to handle open-world queries. The authors should conduct experiments to assess the autoencoder's impact on text embeddings by randomly selecting open-world terms and evaluating the feature correlation before and after encoding.
3. The proposed method requires a set of text queries as input. How are these text queries obtained? Do they need to comprehensively include all objects present in the scene? In the experiments, are the open-vocabulary test queries the same as the input text queries? If so, this could bias comparisons with LangSplat, which does not see the text queries prior to training.
4. Does the proposed method support multi-scale segmentation similar to LangSplat?

**Questions:**

Please see weaknesses.

---

> ### Comment · Reviewer_9XDx · 2024-11-23
> **Concerns not addressed.**
>
> I appreciate the authors' response, but I still have some remaining concerns:
>
> 1. The authors claim that a 5-layer MLP, trained per scene, can compress the embeddings of a large VLM to a six-dimensional representation without any loss of information. This assertion appears counterintuitive. Open-world queries typically have high-dimensional embeddings, yet the proposed 5-layer MLP is trained on a very limited subset of data—the features of objects within a single scene. While it is true that the text and image embeddings are well-aligned before compression, the encoding process performed by the 5-layer MLP could disrupt these correlations. For instance, prior to compression, the text embeddings for 'bird' and 'chair' might be significantly distant from each other, reflecting their semantic disparity. However, after compression, these embeddings might become undesirably close, undermining their original relationships. The authors need to provide evidence that such distortions do not occur; otherwise, the method may fail to effectively handle open-vocabulary queries.
>
> 2. The authors state that text queries are provided at test time. However, line 209 mentions that the method requires a set of text queries 𝑇, which is even used during training to supervise the autoencoder, as indicated in Eq. 1. Furthermore, line 304, 305 reveal that
> 𝑇 is utilized in scene optimization to compute the cross-entropy loss. This clearly demonstrates that 𝑇 is employed during training. Could the authors clarify this apparent contradiction between the claim that text queries are provided at test time and their evident use during training?
>
> 3. The authors claim that the proposed method is capable of performing multi-scale segmentation. However, there are no comparisons or experiments provided to validate this claim. Does the method require retraining to perform multi-scale segmentation? Additionally, the proposed design does not appear to incorporate hierarchical semantics. Could the authors elaborate on the exact technical process through which multi-scale segmentation is achieved?
>
> In summary, I find that the authors' response does not adequately address my concerns. Unless the authors provide additional information or clarification to resolve these issues, I have decided to lower my score.

---

### Meta-Review · Area_Chair_VXTo · 2024-12-20

**Metareview:**

This paper targets 3D open vocabulary semantic segmentation. Different from existing methods, it proposes CRR to refine SAM and CLIP to get more accurate semantic features, moreover, it proposes a low dimension contextual space to reduce the dimension of semantic features so as to improve the computational efficiency. Experiments show that the proposed method achieves state-of-the-art results.

Reviewers acknowledge that the paper is well motivated, the CRR and the low dimension contextual space is novel and thorough experiments demonstrate the effectiveness of the proposed method. As mentioned by several reviewers, the paper lacks some implementation details and the paper needs to be polished for better understanding. Overall, this paper meets the bar for publication at ICLR.

**Additional Comments On Reviewer Discussion:**

Reviewer Mv8f, C9FH and ZWwm all mentioned that the authors addressed their concerns and rate this paper as above the threshold.

Reviewer 9XDx bings up two major concerns: 1. Whether the autoencoder which is used to reduce the dimension of semantic feature impacts performance on open-vocabulary queries; 2. The importance of multi-scale segmentation.
Reviewer 9XDx brings a very good point  that the autoencoder trained per-scene with limited data and labels will impact the performance of open-vocabulary. The authors addressed this issue by explaining the particular setup of 3D open-vocabulary segmentation. All objects appears in the scene are actually "seen" by the autoencoder during training, though their text labels are not used. This explains why the proposed method could perform "open-vocabulary" segmentation with the autoencoder trained with limited data. To support reviewer 9XDx, if the text query during inference is an object never appears in the scene, the behavior of the trained model may be problematic. However, this is not the common evaluation setup of 3D open vocabulary segmentation. The authors follows the setup of prior art to claim "open vocabulary" is not a big concern to me. For the concern of multi-scale segmentation, I agree with reviewer C9FH that this is not a major focus of this paper.

---

### Decision · Program_Chairs · 2025-01-22

Accept (Poster)